November 18, 2022

# Retraction Notice

Retraction: Del Toro I, Ribbons RR. 2020. No Mow May lawns have higher pollinator richness and abundances: An engaged community provides floral resources for pollinators. PeerJ 8:e10021
https://doi.org/10.7717/peerj.10021

After finding several potential inconsistencies in data handling and reporting, the authors and editorial team have agreed to retract this article with the opportunity for re-evaluation should the authors choose to submit a new version.

PeerJ Editorial Office. 2022. Retraction: No Mow May lawns have higher pollinator richness and abundances: An engaged community provides floral resources for pollinators. PeerJ 8:e10021/retraction
https://doi.org/10.7717/peerj-e10021/retraction.

PeerJ Editorial Office. 2022. Retraction: No Mow May lawns have higher pollinator richness and abundances: An engaged community provides floral resources for pollinators. PeerJ 8:e10021/retraction
https://doi.org/10.7717/peerj.10021/retraction



# No Mow May lawns have higher pollinator richness and abundances: An engaged community provides floral resources for pollinators

Israel Del Toro[1] and Relena R. Ribbons[2]

[1] Biology, Lawrence University, Appleton, WI, United States of America
[2] Geosciences, Lawrence University, Appleton, WI, United States of America

## ABSTRACT

No Mow May is a community science initiative popularized in recent years that encourages property owners to limit their lawn mowing practices during the month of May. The goal of No Mow May is to provide early season foraging resources for pollinators that emerge in the spring, especially in urban landscapes when few floral resources are available. We worked with the city council of Appleton, Wisconsin, USA. to allow No Mow May to take place in May 2020. Four hundred and thirty-five property owners registered for No Mow May in Appleton. We measured floral and bee richness and abundance in the yards of a subset of homes ($N = 20$) located near regularly mowed urban parks ($N = 15$) at the end of the month. We found that homes that participated in No Mow May had more diverse and abundant flora than regularly mowed green spaces throughout the city. No Mow May homes had three times higher bee richness and five times higher bee abundances than frequently mowed greenspaces. Using generalized linear models, we found that the best predictor of bee richness was the size of the designated unmowed area, and the best predictors of bee abundances were the size of the unmowed area as well as floral richness. While our findings cannot conclusively attribute increases in bee abundances and richness to the No Mow May efforts, our data does show that bee pollinators make use of no mow spaces as key floral resources during early spring in the upper midwestern United States. A post-No Mow May survey revealed that the participants were keen to increase native floral resources in their yards, increase native bee nesting habitat, reduce mowing intensities, and limit herbicide, pesticide, and fertilizer applications to their lawns. The No Mow May initiative educated an engaged community on best practices to improve the conservation of urban pollinators in future years.

## INTRODUCTION

As landscapes become increasingly urbanized, biodiversity is threatened by land use modifications, a changing climate, and poor management practices (*Elmqvist, Zipperer & Güneralp, 2016*). A notable component of the urban landscape in the United States is

Corresponding author
Israel Del Toro,
israel.deltoro@lawrence.edu,
israedt@gmail.com

a monoculture lawn that is heavily manicured with frequent mowing, and chemically managed. In Wisconsin many lawns are typically seeded with fine fescue or Kentucky bluegrass and in the United States managed lawns account for a land surface area greater than any cultivated crop (*Milesi et al., 2005*). In order to protect as much biodiversity as possible, urban landscapes must be a careful balance of natural habitats, managed urban greenspaces (often consisting of large lawn areas), and functional urban spaces (i.e., spaces that can provide recreational services while maintaining healthy ecosystems) that can accommodate many species (*Shochat et al., 2010*). These urban areas can also be essential for protecting the hundreds of native bee species through supplementing foraging resources for both native and non-native species that provide ecosystem services.

Insects play a large role in a variety of critical ecosystem services that shape and maintain natural and urban landscapes (*IPBES, 2016*), and there is increasing recognition that their conservation is vital in light of trends of global insect declines. These ecosystem services include provisioning, cultural, supporting, and regulating services ranging from nutrient cycling to pollination (*Prather et al., 2013*; *Noriega et al., 2018*). One functional group of interest for protection are native pollinators which are integral to sustaining agricultural food systems (*IPBES, 2016*) and may play important functional roles in urban settings (*Hall et al., 2017*). Urban and suburban landscapes have the potential to protect and enhance wild bee diversity and abundances (*Pardee & Philpott, 2014*; *Baldock et al., 2015*; *Baldock et al., 2019*; *Lowenstein, Matteson & Minor, 2015*; *Wilson & Jamieson, 2019*; *Wenzel et al., 2020*) via careful policy development (*Hall & Steiner, 2019*) and promotion of pollinator friendly behaviors among the urban public (*Hall et al., 2017*; *Zattara & Aizen, 2019*; *Cardoso et al., 2020*).

The state of Wisconsin lists nearly 500 species of native bees (*Wolf & Ascher, 2008*). In the city of Appleton, we have previously documented 89 species of wild bees in urban green spaces and suburban nature reserves (Anderson et al., 2017–2018, unpublished data). Some of these are early emerging species, coming out of winter hibernation between late April and early June, as temperatures go above freezing and daylength increases in Northeast Wisconsin. During this time, there may be limited forage available, especially in fairly homogenous mowed urban lawn environments, where herbaceous vegetation is not given enough time to flower. The flora in these lawn areas may provide abundant forage for urban wild bees (*MacIvor, Cabral & Packer, 2014*).

The displacement of native wildflower and tree forage by lawns has removed a vital early season nectar and pollen resource for many pollinators, including bees. One initiative that was popularized in the United Kingdom through the organization *Plantlife (2020)*, aimed at allowing flowers to bloom in lawns throughout the month of May to provide the floral nectar needed for pollinators. This initiative has been dubbed "No Mow May" and led researchers to follow up with an "Every Flower Counts" community initiative to document which flowers were common to their blooming lawns. Additionally, previous work has shown that reducing mowing intensity will have positive impacts on urban bee abundance and diversity (*Lerman et al., 2018*, *Wastian et al. 2016*) but it remains unclear how generalizable these results are. Other initiatives that promote the creation of bee habitat in urban landscapes include the "Lawns to Legumes" project Minnesota

(https://bwsr.state.mn.us/l2l) and the Xerces Society's and National Pollinator Network's "Million Pollinator Garden Project" (http://millionpollinatorgardens.org/). These national efforts are also complemented by more localized efforts with similar pollinator protection project like the "Appleton Pollinator Project" (http://www.BYOBEEZ.org).

The goal of No Mow May in Appleton, Wisconsin was to increase the floral forage resources critical for early emerging pollinator species. A second goal of this initiative was the outreach and education regarding the protection of native wild pollinators in urban and suburban settings. Our main objective was to test whether not mowing during the month of May had an effect on bee richness and abundances in the city of Appleton WI, USA. We did this by comparing bee richness and abundances in No Mow May participant's lawns, relative to regularly mowed parks in the city. We also aimed to document the diversity of floral resources in both the mowed parks and the unmowed lawns as these are likely the resources that bee pollinators are using in urban yards. Our final objective was to document community perspectives after participating in the No Mow May initiative to see how landowners plan to manage lawns in the future and enhance pollinator friendly practices.

## MATERIALS & METHODS

### Study area

Appleton, Wisconsin is located at the transition zone between Northcentral Hardwood Forests and Central Wisconsin Till (Environmental Protections Agency Ecoregion Level III) within the upper Midwestern United States. The city has a population of 74,098 with upwards of 30,000 households (US Census estimates July 2019). The average low temperature in Appleton for the month of May was 1.4 °C and the average high was 12.1 °C (National Weather Service). We selected 20 homes in five neighborhoods of the city of Appleton (presented as Fig. 1) to sample bee richness and abundance during the final week of No Mow May (May 25 to May 30, 2020), with the help of eight volunteers. Homes were selected based on being unmowed through the entire month, and their proximity (within 1 km) to the focal parks where bee diversity was also evaluated. Whenever possible, homes were separated by at least 100 m to reduce the likelihood of resampling individual bees at multiple homes.

The No Mow May initiative in the city of Appleton consisted of 435 registered participants (Fig. 1), or ~1.5% of Appleton residences. There was also participation in the city by many unregistered participants, but we were not able to quantify what percentage of the city did not register yet still participated in No Mow May. Of the 435 registered participants, 130 responded to the post-No Mow May survey, a ~30% response rate. At the subset of 20 homes the mean unmowed area was 195 sq. meters, ranging from 91 sq. meters to 446 sq. meters.

### Working with city government and the community

The No Mow May efforts in the city of Appleton WI USA (Fig. 1) required the approval and close collaboration with the city government and city residents. The city has strict guidelines on lawn care practices including a residential 20 cm (8 inch) allowed maximum lawn

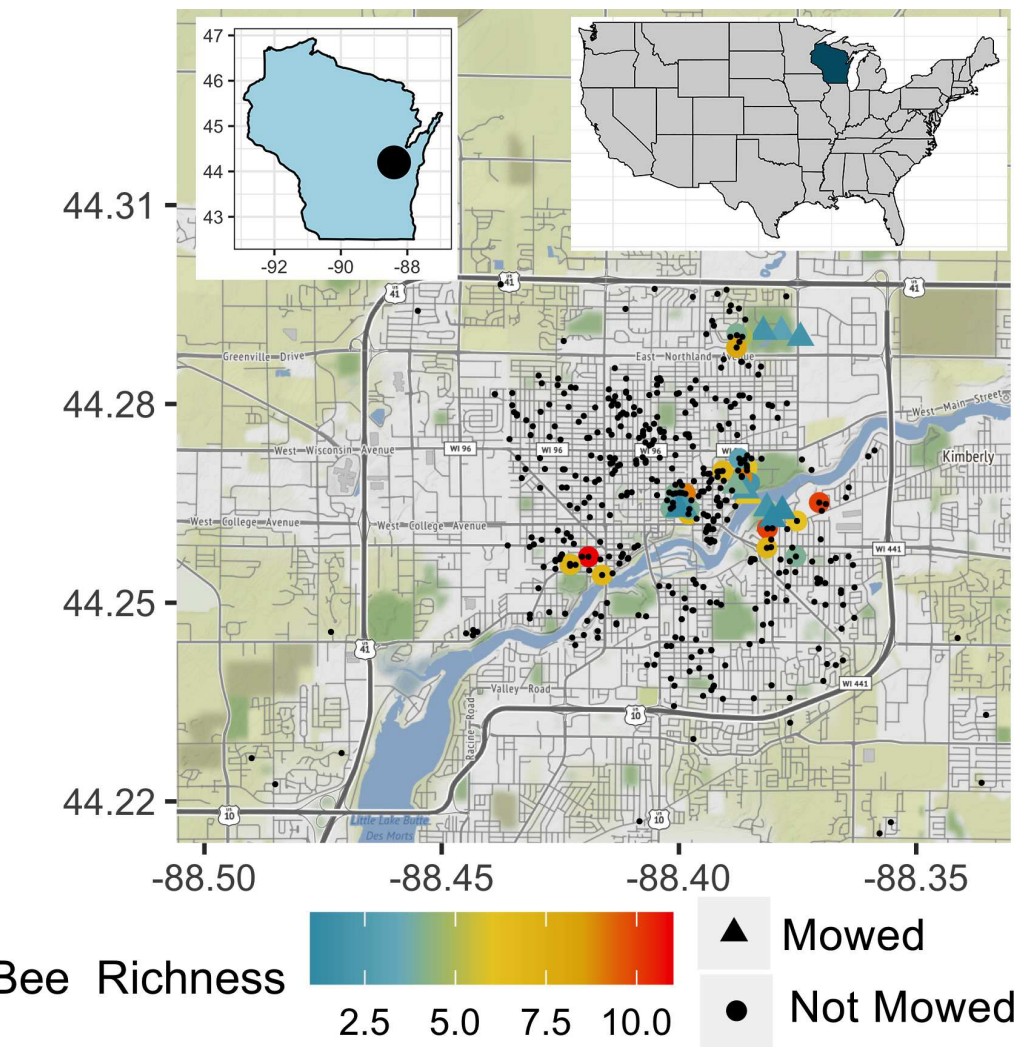

**Figure 1 Map of the study area.** Map of the city of Appleton showing participating homes in No Mow May (black points) created in ggmap (*Kahle & Wickham, 2013*). Color points indicate lawns and colored triangles show green spaces where bee diversity and abundance were recorded. Insets show the location of Appleton in the state of Wisconsin and a national map of states with Wisconsin colored in.

height in residential properties and 31 cm (12 inch) maximum lawn height in commercial properties. Local government officials petitioned the city to waive the ordinance for the month of May 2020. After multiple meetings, discussion with city officials, and a vote in the city common council the resolution was approved in April 2020. Community members of the city of Appleton were asked to register their homes as participants of No Mow May via an online form. A local pollinator advocacy group, The Pollenablers Fox Cities, worked on outreach and education to inform the community of Appleton of the agreed upon rules and regulations of the No Mow May initiative via instructional videos, social media, and printed materials. Many of these materials can be found on the Appleton Pollinator Project website (http://www.BYOBEEZ.org).

## Data collection

We used standardized timed sweep netting as our method of bee collection. At each home we measured the area designated by the resident as a unmowed area and standardized our sampling based on square meters. For each sampling location, we standardized sampling by dedicating one-person hour of sampling per 200 square meters of unmowed area. Sampling was completed only during fair weathered days when air temperatures ranged from 21 °C to 27 °C, mostly sunny and clear skies and low wind speeds <8 kph (there was one day of rain which prohibited sampling). As we netted suspected bee specimens, the bees were moved into storage mason jars. Collected bees were identified in the field using a well-established and verified reference collection for the city of Appleton (obtained from (Anderson et al., 2017-2018, unpublished data)), then released once the sampling period was concluded at each site. Unknown specimens were stored in 70% ETOH, and taken to the laboratory for subsequent identification using various keys and regional lists (*Wolf & Ascher, 2008*; *Williams et al., 2014*; *Wilson & Carril, 2015*; *Gibbs et al., 2017*).

At each home we compiled a flowering vegetation species list, and measured percent cover in five 1-square meter plots of herbaceous vegetation relative to lawn grasses or bare ground. We selected the vegetation plot locations randomly after gridding the total lawn area and using random number generators to designate percent cover plot locations within that grid. For subsequent analyses we used the mean of these five vegetation assessments as a predictor of bee richness and abundance. Plants were identified to species through personal knowledge and using University of Wisconsin herbarium keys (University of Wisconsin Herbarium, 2020) where necessary. We also sampled 15 mowed areas in the city of Appleton, intended to serve as a standardized mowed plot comparison for lawns. The city of Appleton manages these parks by mowing every 5 to 7 days, applying vinegar-based solution as an herbicide and unspecified fertilizers. Individual lawns that were regularly mowed would be a useful comparison, but given the opt-in nature of this experiment, individuals who did not express interest in the program (regardless of ability to participate) were not numerous enough to serve as a sufficient standardized control. At each park we sampled 150 square meters for a total duration of 45 min to remain consistent with our sampling methods of residential yards. All park plots were a minimum of 100 m apart from each other to help reduce the probability of recapturing bee specimens in multiple plots. We acknowledge that for some of the larger species, foraging ranges exceed 100 m and every effort was made to avoid resampling and recapturing sampled individuals. While park plots and lawns do have distinctive differences in function and use, we used parks due to logistics of acquiring approval for sampling and the consistency of lawn care practices applied to all parks throughout the city.

## Post no mow may survey

Immediately after completing the month of May sampling, we surveyed all 435 registered participants regarding their perceptions of the results of No Mow May and how their lawn care practices might change, and 130 of the participants responded. We asked participants about perceptions of pollinators and flowers in their yards: (1) Did you see pollinators in your yard this year? and (2) Did you see more flowers in your yard this year? We also

**Table 1 List of the most common flowering plants in home and park lawns.**

| Name | Common Name | Percent of homes present | Percent of parks present |
|---|---|---|---|
| *Taraxacum officinale* | Dandelions | 100% | 73% |
| *Viola papilionacea* | Violet | 95% | 20% |
| *Trifolium repens* | White Clover | 80% | 60% |
| *Glechoma hederacea* | Creeping Charlie | 75% | 13% |
| *Capsella bursa-pastoris* | Shepherd's Purse | 75% | 40% |
| *Plantago major* | Plantain | 70% | 53% |
| *Cirsium arvense* | Canada Thistle | 30% | 7% |

asked participants how their mowing habits might change as a result of No Mow May and offered a checklist of things they could do in their yards to help local pollinators. The Lawrence University Institutional Review Board (IRB: 5_10Del Toro) approved this questionnaire and all responses were kept fully anonymous and confidential. No identifying or demographic information was collected. Informed consent was obtained from participants and IRB details were shared with participants when they completed the online form.

## Data analyses

All analyses and plots were completed using the R statistical software v. 4.0.0, ''Arbor Day'' (*R Development Core Team R, 2014*). We compared the medians of observed bee richness (the total number of species present in a given site) and abundance in mowed and unmowed lawns using a Kruskal Wallis comparison of means. We used this non-parametric alternative due to relatively low sample sizes and variation in the normality of the data. We then used floral richness, percent cover of herbaceous flowering vegetation, and size of the sampling area as predictors of bee richness and abundance in a generalized linear model (glm) assuming a Poisson family distribution, which is required for count data. The glm was simplified using step-wise variable selection using the function ''stepAIC'' in the MASS package (*Venables & Ripley, 2002*), based on the Akaike Information Criterion (AIC).

## RESULTS

### Floral and bee diversity

The most common and abundant floral resources in lawns and greenspaces are reported in Table 1, with *Taraxacum officinale,* the common dandelion, present in all home lawns and at 73% of urban park lawns, making it the most abundant plant and floral resource in lawns (Table 1). Floral richness and abundance were higher in unmowed lawns relative to mowed greenspaces (Kruskal-Wallis chi-squared $= 14.49$, $df = 1$, $p = 0.0001$ for Richness, Kruskal-Wallis chi-squared $= 16.82$, $df = 1$, $p = 0.000004$ for floral density). Mowed Areas had 36% fewer plant species and 34% lower flower density than not mowed areas (Fig. 2). The complete floral species list can be found in the supplementary material.

We collected a total of 321 bees, consisting of 33 bee species, during the week of intensive sampling. The five most abundant species were *Lasioglossum cressoni, Hoplitis pilosifrons,*

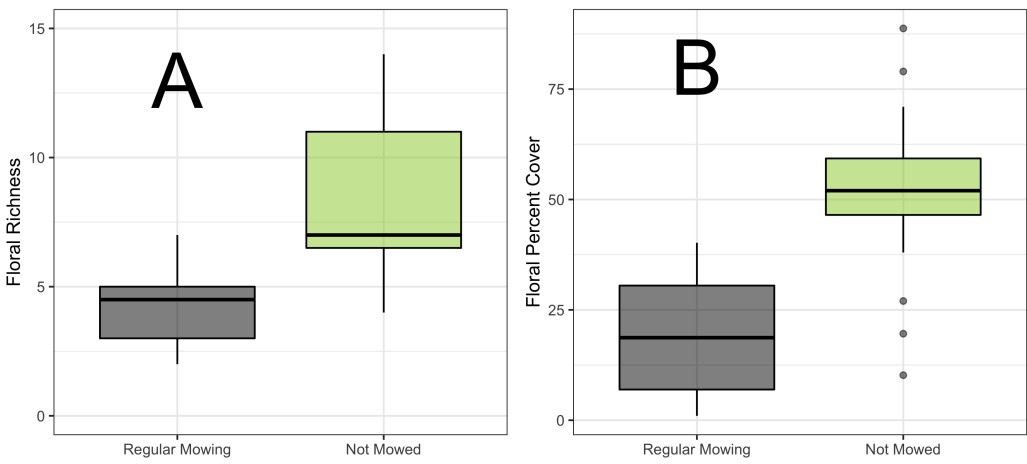

**Figure 2** **Boxplot of floral richness and percent cover comparisons.** Boxplot showing higher median floral density (A) and richness (B) in No Mow May lawns ($n = 20$) relative to regularly mowed areas ($n = 15$).

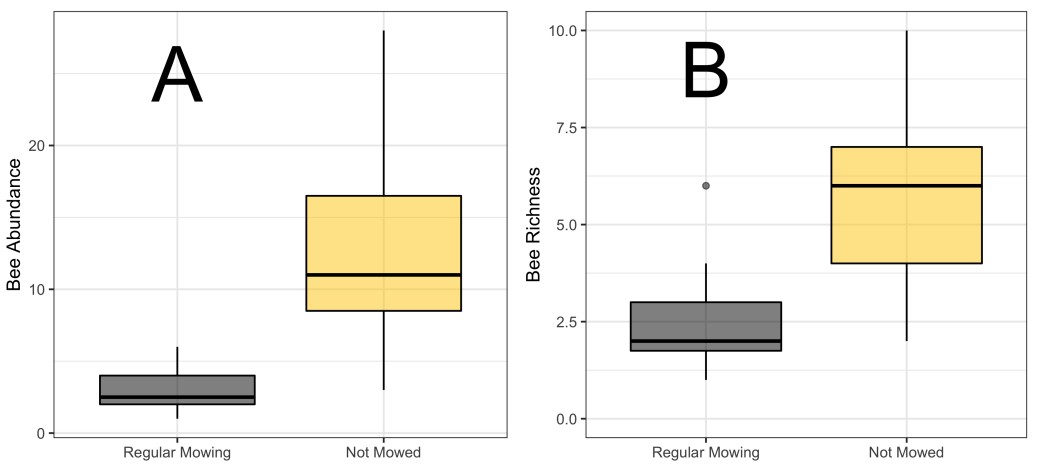

**Figure 3** **Boxplot of bee abundance and richness.** Boxplot showing higher median bee abundance (A) and richness (B) in No Mow May lawns ($n = 20$) relative to regularly mowed areas ($n = 15$).

*Melissodes bimaculatus, Apis mellifera,* and *Bombus impatiens* which accounted for 65% of all observed individuals. Bee abundances and richness were higher in unmowed lawns relative to the regularly mowed green spaces (Kruskal-Wallis chi-squared = 19.72, $df = 1$, $p = 0.00000006$ for bee abundance and Kruskal-Wallis chi-squared = 16.69, $df = 1$, $p = 0.00004$ for bee richness). Median bee abundances were nearly five times higher and bee richness was three times higher in unmowed lawns relative to regularly mowed plots (Fig. 3, Table 2).

Generalized linear model results show that bee abundance is best predicted by the additive effects of total area that remained unmowed and the floral richness at each lawn (AIC of reduced model = 199.51, Null Deviance = 170 on 31 Degrees of Freedom.
**Table 2   List of bee species and abundances collected in this study.**

| Family | Species | Count in mowed sites | Count in unmowed sites |
|---|---|---|---|
| Andrenidae | *Andrena crataegi* | 4 | 8 |
| | *Andrena cressoni* | 3 | 8 |
| | *Andrena miranda* | 0 | 6 |
| | *Andrena wilkella* | 0 | 1 |
| Apidae | *Apis mellifera* | 1 | 17 |
| | *Bombus impatiens* | 2 | 15 |
| | *Bombus rufocinctus* | 0 | 1 |
| | *Bombus vagans* | 1 | 6 |
| | *Ceratina calcarata* | 0 | 4 |
| | *Melissodes bimaculatus* | 3 | 16 |
| | *Melissodes denticulatus* | 2 | 0 |
| | *Melissodes desponsus* | 0 | 1 |
| | *Melissodes druinellus* | 2 | 1 |
| | *Melissodes rustica* | 1 | 1 |
| | *Nomada cressoni* | 0 | 7 |
| Halictidae | *Agapostemon virescens* | 3 | 7 |
| | *Augochlorella aurata* | 0 | 3 |
| | *Augochlorella pura* | 2 | 5 |
| | *Halictus ligatus* | 0 | 6 |
| | *Halictus rubicundus* | 0 | 1 |
| | *Hoplitis pilosifrons* | 12 | 22 |
| | *Hylaeus modestus* | 0 | 2 |
| | *Hylaeus mesillae* | 0 | 3 |
| | *Lasioglossum coriaceum* | 1 | 4 |
| | *Lasioglossum cressonii* | 22 | 98 |
| | *Lasioglossum laevissimum* | 1 | 5 |
| | *Lasioglossum pilosum* | 0 | 1 |
| | *Lasioglossum zephyrum* | 1 | 2 |
| | *Sphecodes cressoni* | 1 | 1 |
| | *Sphecodes dichrous* | 0 | 1 |
| Megachilidae | *Anthidium manicatum* | 0 | 1 |
| | *Megachile campanulae* | 1 | 0 |
| | *Osmia pumila* | 0 | 4 |

However, bee richness was only best predicted by the effect of the total area that was not mowed (AIC of the reduced model = 144.69, Null Deviance = 56.48 on 32 Degrees of Freedom). The full stepwise model summary is available in the supplementary material. Floral density, while significantly higher in unmowed lawns, did not have a significant effect on bee abundance or richness.

## Post-no mow may survey results

Based on the two perception questions that we asked participants (1) Did you see pollinators in your yard this year? and (2) Did you see more flowers in your yard this year? About 60% of respondents noticed a few more or a lot more pollinators and flowers in their lawns this year, and about 20% noticed no change or fewer pollinators and flowers than normal in their yards during the month of May.

We also asked participants how they might modify their lawn management practices. 77% of respondents pledged to reduce or eliminate the use of chemical herbicides or pesticides in their lawns, 62% pledged to reduce or eliminate the use of chemical fertilizers in their lawns, 57% planned to increase native pollinator habitat in their yards and lawns, and 48% planned to plant native floral resources as forage for pollinators. Eighty seven percent of participants said they would participate in No Mow May again in future years.

## DISCUSSION

Urban environments can provide opportunities for promoting floral resources for pollinator conservation, which are especially important for early emerging bee species during a time of year when food resources may be scarce. Our findings are consistent with *Lerman & Milam (2016)* who documented bee abundance in suburban landscapes in Western Massachusetts and suggested that spontaneous lawn flowers offer supplemental floral resources that can support pollinators. Lawns can provide important food sources that promote healthy pollinator populations in urban ecosystems, if managed intentionally. The data we provide here adds to the growing body of literature that urbanized landscapes can provide sufficient forage for wild bees and enhance bee diversity and abundances (*Fetridge, Ascher & Langellotto, 2008*; *Pardee & Philpott, 2014*; *Lowenstein, Matteson & Minor, 2015*; *Hall et al., 2017*; *Baldock et al., 2019*). We suspect that as more floral resources become available towards the end of spring and into summer, wild urban bees will transition to using additional foraging resources which may not be common species in traditional lawn or grassy areas, however this pattern remains to be documented.

No Mow May lawns have a fivefold higher bee abundance and threefold higher bee species richness compared with regularly mowed areas. This is the first study, to our knowledge, to document the specific observed effects of No Mow May practices on bee abundances and richness. Previous studies have detailed that different mowing practices will impact the diversity and abundances of insects (*AndreasUnterweger, Rieger & Betz, 2017*) including bees (*Lerman & Milam, 2016*; *Lerman et al., 2018*). Generally, higher mowing intensity is negatively associated with decreased abundances and diversity. Our rapid assessment offers support for the same effect of mowing practices during early spring in the Upper Midwestern USA on urban bee diversity and abundance. We found that the area of lawn that was not mowed was a key predictor in both bee abundance and richness while plant species richness only helped to explain bee species richness (glm results). The positive relationships between increased area corresponding to increased species abundance and richness, (i.e., "Species Area Relationship"), are well documented in the ecological literature (*Dengler, 2009*) and seem to apply to the patterns detected in urban

ecosystems as well (*Matthies et al., 2017*). The pattern of increased plant richness correlating with increased bee richness is consistent with the ecological patterns of "diversity begets diversity" where increased richness in one taxonomic group promotes richness in a closely associated group (e.g., when habitable or usable space is limited *Maynard et al., 2017*). From an applied perspective, if clusters of neighbors were to participate in No Mow May initiatives then bee species richness and abundance should consequently increase in these yards as a result of having a larger undisturbed contiguous area. The positive effect of plant species richness on bee species richness is consistent with the more heterogenous and diverse landscapes tending to provide increased niche space for hosting more species, another well documented ecological pattern (*Ebeling et al., 2008*; *Abbate et al., 2019*).

Bees are amongst the key insect groups that provide essential ecosystem services (*IPBES, 2016*; *Noriega et al., 2018*). While the agricultural value of bee's providing pollination ecosystem services has been thoroughly explored (*Hanley et al., 2015*), their roles in providing these services in urban ecosystems remains poorly understood. It likely that bees also provide a valuable pollination ecosystem service in urban landscapes (*Normandin et al., 2017*). Previous work has shown that if the conservation goal is protection of species (and consequently the ecosystem services they provide), then cities are likely to play essential roles as they can be home to as many if not more species than "natural" habitats (*Baldock et al., 2015*).

The No Mow May initiative in the city of Appleton went beyond the reduction of mowing practices in the community. This initiative also started a community-wide discussion on best practices for pollinator conservation. Even though not all community members were participants in No Mow May, this city-wide initiative offered educational opportunities through social media platforms (http://www.facebook.com/pollenablers), traditional media interviews (television, radio and newspaper) and by word of mouth on the benefits of transforming lawns into pockets of urban habitat that can support and harbor native biodiversity. We aim to more accurately quantify how these educational efforts are being received by the city of Appleton in future years. We promoted best practices that have positive effects on our pollinator communities like the planting of native wildflowers (*Pardee & Philpott, 2014*), increasing wild bee nesting habitat (*Harmon-Threatt, 2020*) and reducing herbicide and pesticide use (*Muratet & Fontaine, 2015*; *Aronson et al., 2017*). As this initiative grows in the city, we aim to expand our education and outreach efforts so that a broader audience is reached and hopefully motivated to participate. Although we did not evaluate how widespread these practices are in Appleton, some of the city has now been exposed to educational opportunities needed to promote a more sustainable and pollinator friendly community. In general, communities tend to be aware of the importance of bees in urban ecosystems but lack education on how to better protect them (*Wilson, Forister & Carril, 2017*), or ways to participate in community initiatives (*Bloom and Crowder, 2020*).

In a "snapshot" study of this nature, the role of community involvement and buy-in was essential. From a study design perspective, we had the capacity to choose our sampling locations from over 400 sites around the city, allowing for a robust, standardized, and systematic sampling design. Due to logistical constraints, and the necessity for rapid inventory, we subsampled from five neighborhoods around the city. Our study enhanced

awareness of key ecological and conservation issues, improved the general public's understanding of urban ecosystems, provided community members the opportunity to participate in data collection all of which are common individual and programmatic outcomes of any community science project. We hope that our efforts have also enhanced trust and communication between the general public and the local scientific community which is can be a desirable community-level outcome of a project like No Mow May (*Jordan, Ballard & Phillips, 2012*). We anticipate that, with the resulting data, community involvement in development of pollinator protection policy at the city and regional level is a likely future direction, which is also a valuable outcome of community science (*Adler, Green & Şekercioğlu, 2020*). As an example, alderpersons are currently drafting city ordinance proposals to enhance pollinator habitat along community trails, new housing developments, electric line corridors and water management retention ponds. As many community science projects can attest, communities are interested in education and participation in the scientific process. No Mow May is an initiative that exemplifies the adaptability and interest of landowners in moving towards conservation practices that promote healthier and more resilient ecosystems.

Lawns are easily accessed urban spaces that can serve to protect native biodiversity. We suspect that for a city the size of Appleton (65 square kilometers) at least 100 acres (40 hectares) of lawn area can be managed to provide early season forage for native pollinators by engaging in initiatives like No Mow May. The notable higher abundance and richness in unmowed areas suggests that the very least, the resulting floral resources are attracting urban bees. Longitudinal studies are needed to track the temporal abundances of populations as our communities transform into more pollinator friendly landscapes. No Mow May might not be suitable for all urban ecosystems as much of North America enters the spring season earlier that the Upper Midwest, and thus this initiative might be better as No Mow March or No Mow April in warmer parts of the country. We also recognize that this rapid biodiversity assessment is a snapshot of what occurs seasonally in urban ecosystems. Describing the diversity patterns of bees in urban settings is a complex ecological story. While the effect of not mowing during the month of May is documented here, there are additional drivers (e.g., regional floral diversity, access to water and proper nesting substrate, lawn mowing frequency and intensity) and correlates of bee richness and abundances that should be carefully explored in future studies. Ongoing work seeks to investigate the longitudinal patterns of urban bee diversity in this region of the United States as well as a more detailed understanding of environmental attributes of urban greenspaces that can enhance pollinator activity and diversity (Anderson et al., 2017–2018, unpublished data). We aim to continue our sampling and outreach and education efforts by expanding this effort to the entire Fox Cities Region in 2021 and promote a state-wide No Mow May effort in subsequent years.

## CONCLUSIONS

The effect of our No Mow May effort documented increases in both urban bee and floral abundances and diversity. We found that the area that remains not mowed was

the strongest predictor of bee abundance and diversity, while floral species richness also contributed to explaining bee species diversity in mowed and unmowed areas in the city of Appleton. Based on our survey results, we found strong community enthusiasm regarding this initiative with the majority willing to continue this and other pollinator friendly practices in their homes and neighborhoods. In order to ensure that lawns can maximize pollinator biodiversity protection, and valuable cultural ecosystem services then we have to think critically about new norms for lawn maintenance which are more effective when implemented at the neighborhood and community levels (*Nassauer, Wang & Dayrell, 2009*). In our case, Appleton city residents' perceptions are often that well-manicured and low "weed" diversity lawns are preferable. However, from a conservation and ecological perspective these types of lawns may not be in line with the community biodiversity conservation values. One way to overcome this issue is by increasing community outreach and awareness about the importance of protecting urban bees and providing important foraging resources for them. Additionally, future research directions should aim to explore the city's aesthetic perceptions of mowed lawns relative to unmowed lawn alternatives, because this information can help shift public opinion, reinforce existing positive perceptions of flowering lawns and address the city's concerns about pollinator protection initiatives (*Ramer et al., 2019*).

## ACKNOWLEDGEMENTS

We would like to thank the city of Appleton for allowing the No Mow May initiative, especially Alderpersons Schultz, Metzler, Martin, and Fenton for their support in writing the ordinance that allowed the community of Appleton to grow out their lawns. We also thank the following volunteers that contributed by helping sample neighbor's lawns and urban green spaces, Joan Ribbons, Beth Johnstone, Kim Grummer, Mario Seaman, Reiko Ramos, and 3 anonymous volunteers. We thank all Appletonians who participated in No Mow May, their efforts and interest has generated an informed community with a keen focus on pollinator protection. We appreciate the thorough input of Douglas Martin on previous drafts of this work. We also acknowledge that the city of Appleton is registered with the Xerces Society as a member of the Bee City USA program.

### Funding

The authors received no funding for this work.

### Competing Interests

The authors declare there are no competing interests.

### Author Contributions

- Israel Del Toro conceived and designed the experiments, performed the experiments, analyzed the data, prepared figures and/or tables, authored or reviewed drafts of the paper, and approved the final draft.

- Relena R. Ribbons conceived and designed the experiments, performed the experiments, authored or reviewed drafts of the paper, and approved the final draft.

## Ethics

The following information was supplied relating to ethical approvals (i.e., approving body and any reference numbers):

The Lawrence University Institutional Review Board granted ethical approval to carry out the citizen surveys (IRB: 5_10_20_deltoro).

## Data Availability

The data are available in the Supplementary Files.

## Supplemental Information

Supplemental information for this article can be found online at http://dx.doi.org/10.7717/peerj.10021#supplemental-information.

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
