# Peer review of "Retraction Notice"

_PeerJ, doi:10.7717/peerj.10021_

## Round 0.1 · original submission · Major Revisions

Thank you for your manuscript. Three reviewers and I have read it and we agree that while the paper is a worthwhile contribution, the reporting needs to be revised.

Each of the three reviewers has left detailed comments. Please address these in a follow-up revision.

Thank you again for your submission and I look forward to a revised version.

·

Basic reporting

I suggest a number of additional references to consider. I also recommend some slight tweaks to Table 2 and Figure 1.

Experimental design

Research question well defined though framed as objectives. The authors could rephrase as hypotheses if that is the standard for PeerJ. The authors could be more explicit about how they fill research gaps.

I had concerns with the study design, namely the comparison of mowed parks with no mow lawns. See below. Some of the method section requires revising and more clarity. I also am concerned about the lack of independence amongst sites.

Validity of the findings

For the most part, the findings are solid. However, the discussion does get a bit too speculative, particularly because of the small sample size and the comparisons of two different land use types. I suggest the authors address the limitations of their study (which doesn't take away from the novelty).

Additional comments

The manuscript titled “No Mow May lawns have higher pollinator richness and abundances: An engaged community of citizen scientists protecting pollinators and their floral resources” tests how bees respond to a city-wide program aimed at providing pollinator habitat. The study also included a survey to assess perceptions of biodiversity in yards. Providing the scientific evidence to support programs that aim to enhance habitat are important and can highlight the strengths as well as areas of improvement for these types of initiatives. I think this is an important study and has implications for conservation. My main concerns revolve around the site selection, i.e., comparing No Mow yards with mowed parks since these are two different types of green spaces, managed with different objectives, and different sizes and configurations, all of which can influence bees. I think having a lot more caution when discussing the results, in combination with highlighting the study limitations could help ensure the interpretations don’t go beyond the study. I also was not clear about the role of the ‘citizen scientist’, and although they helped collect data, I don’t think this was a citizen science project. If I’m mistaken, then the method section requires more detail. And I think the manuscript could be a bit more explicit about what research gaps the study fills. Nonetheless, I enjoyed reading the manuscript, was excited about the results, and see several future research opportunities based on this study system.

Specific comments (with full citations at the end):

Title
I don’t think the strongest aspect of the study is the role of the citizen scientists, and I suggest changing the title that better reflects how households embraced a city-wide program to provide pollinator habitat. I don’t think the study assesses whether these lawns ‘protect’ pollinators.
Abstract
L16: I’m more familiar with the term ‘citizen science’, or to be more inclusive, ‘community science’. I suggest using one of these more familiar terms.
L19: switch ‘where’ to ‘when’ since there are lots of floral resources in urban areas, though most likely not as abundant early in the season.
Introduction
L 46-51: I appreciate the inclusion of the different types of urban green spaces. However, I would argue that all these spaces are managed but to a different degree. What is meant by ‘functional urban spaces’? Are these for people as well? And I’m not sure the purpose of calling out Apis mellifera. Native bees also provide key ecosystem services (as well as several other flora and fauna). I think this last sentence can be broader. That will lead into the next paragraph.
L59: I’m not sure the need for ‘pollinator’ when describing wild bees.
L66: Remove ‘the’ before ‘late April’.
L63-71: I think this paragraph can be more general to urban bee studies, and not mention Wisconsin. For example, several studies have published bee lists (e.g., Fetridge et al. 2008, Lerman and Milam 2016). I suggest providing a broader overview of urban bee studies, with a focus on those conducted in yards, different management practices, and how floral abundance influences bee patterns. Additional studies to review: Baldock et al. 2015 and 2019, Frankie et al. 2005, Lowenstein et al. 2015, and Pardee and Philpot 2014.
L72-80: I like how this paragraph describes the No Mow May program. I think this paragraph can be strengthened by briefly describing other programs and management practices that promote pollinator habitat (e.g. Xerces society million pollinator garden project, mow less frequently, the Minnesota Bee Lawn program) to demonstrate other efforts for managing for pollinators in built up areas.
L80: This should reference Lerman et al. 2018, not Lerman and Milam 2016.
L81-83: I think the description of No Mow May would work better in the previous paragraph that describes the program.
L85-87: This is a bit awkward. Were the comparisons made in yards of those participating in the No Mow Initiative? If so, could rephrase to: We compared bee richness and abundance in lawns participating in the No Mow May initiative with lawns regularly mowed. Also, although I can appreciate the challenges of accessing private properties (and public parks) for conducting research, I’m concerned that the study does not compare No Mow yards with ‘Mow yards’. See comments in the Materials and Methods section.
Materials and Methods
L95-106: I like the inclusion of the background of the program. However, I think this first paragraph can be broader to provide more context on study area (i.e., Appleton), that includes information about population size, climate, etc. You can also include information about number of bees recorded if you agree with my suggestion to remove that information from the introduction. Then you can describe the No Mow program, and how many sites you sampled. The next paragraph gets into the details of the data collection. The subheading could be ‘study area’, similar to many other method section first paragraphs.
L109-111: move to above paragraph so this section is all about what data you collected. And who are the 8 citizen scientists? Are these people who volunteered their lawns for the initiative? A subset of the sites? I’m not sure the purpose of mentioning the citizen scientists and what they did for the study.
L108-137: Separate these paragraphs so the first describes the bee sampling and the second describes the floral sampling. I suggest this order since I think the main objective is to test whether bees respond to the different treatments. If they do, then you predict it is because of the floral resources. This order makes more logical sense (to me). They get a bit mixed in the first paragraph (e.g., for L119, is the 45 minutes regarding the sweepnet sampling or was the floral sampling timed?)
L111-113: It is unclear exactly how you measured the floral richness. Did you measure percent cover for each species within the 1-square meter sample? Or is it a percentage of all the flowers in the 1-square meter compared to the area without flowers or with bare soil? And does the mean represent the percent floral cover, percent lawn cover and percent bare lawn of these five 1-meter samples?
L115-123: I’m concerned about independence of your samples. 100 meters might be suitable for small-bodied bees but not for Bombus spp, which can forage / travel 2km+. I suggest accounting for spatial autocorrelation to test whether the 3 sites within each park (and I’m assuming there were three 100-m plots within each of the 5 parks…) were independent. Also, to truly get at the effects of No Mow May, I think would require sampling private yards participating and private yards not participating. Comparing yards with parks invites additional variation into the study design (e.g., different motivations for management, different configurations of green space / fragmentation). Thus, observed differences between parks and yards might be due to other factors besides participating in the initiative. Please justify the study design and why it was not possible to either only sample parks (did some participate in the No Mow May program?) or yards, or, sample no mow in parks and yards, and compare with mowed parks and yards.
L124-137: Can you provide some references to support your methodology? Also, which keys were used?
L136-137: see comment above about 100 meters between sites.
L140-142: Was the survey administered to all households participating in No Mow May or only to the 20 yards? Oh, I see on L192 you provide these details. This is part of the methods and should be moved to the survey section.
L139-148: Including the survey really strengthens this study since it provides more context about who participates in these programs as well as highlights opportunities for improvement. I suggest rephrasing the questions to better reflect the questions. As written, they imply (to me) a dichotomous answer though the way the questions were drafted provides a lot more information. This could be rephrased as something like: we asked participants about perceptions of pollinators and flowers in their yards. Alternatively, you can list the questions and responses verbatim from the survey. I like the phrasing of the mowing habits. Also, check out Ramer et al. 2019 and Ramer and Nelson 2020.
L151: I’m not sure what ‘plots’ refers to in this context.

Results
L164: I think this subheading is misleading given the sample sizes. I think ‘Bee diversity’ is more appropriate.
L169: I’m not sure the formatting rules for PeerJ, but other journals will not write out the exact p-value but rather state: p<0.0001, p<0.001, and p<0.01, depending on how significant.
L174-180: I suggest starting with how many flower species identified, which were the most abundant (similar to the bee results). Then report the stats. And I would describe the patterns in terms of the no mow lawns, again, similar to the bee results (i.e., no mow lawns had XX% more flowers…).
L182: Is there a more affirmative word to use rather than ‘suggest’? And can you provide some stats for these statements?
L188-196: I’m not sure much of this paragraph is ‘results’, but rather provides more context for the study system and how you conducted your sampling. For example, didn’t you need to calculate the no mow area prior to sampling bees? I think L194-196 is very interesting and perhaps more suitable for the Discussion. However, 22-40 acres doesn’t mean much to me without any sort of context. How many acres of mowed lawns are there in Appleton?
L198-209: This paragraph would benefit from a different subheading as the use of ‘citizen’ is confusing. Are these the citizen scientists? Or households participating in the No Mow May program? I suggest the latter. If you agree, then please change throughout. If the questions are written out in the methods (as per my comment above), then no need to repeat. Rather, I suggest simplifying and present the results.

Discussion
I think you can strengthen your Discussion sections by starting with the main findings of your study (much of the second paragraph), and how they relate to other studies. This is the big picture paragraph. Then for subsequent paragraphs, get into more details as to why you saw your results, and their significance.
L239-246: I’m not sure the point of this paragraph and how it relates to the study. Yes, bees are important, and cities can support bees. Can you relate to a specific finding? Otherwise, I suggest deleting.
L247-252: I suggest a more cautious approach when describing (human) community responses to the No Mow program. According to Google, there are 74,526 people in Appleton (I suggest finding the figure from the US Census…). Yes, it is impressive that 400+ people registered for the program, and it is likely this number will increase in future as more people become aware of the program and the program gains broader acceptance. However, for this study, you did not assess the effectiveness of the promotion of the program. Only 600 people follow the facebook page (also impressive!), and so the educational opportunities are limited to a small subset of people. There is a lot of potential and I suggest highlighting that and how the program can build upon the enthusiasm and the ecological results.
L265-268: I think this is a liberal definition of ‘citizen scientist’. Are the no mowers really scientists or are they participating in conservation? I’ve collected data in hundreds of yards across the country yet I wouldn’t refer to the householders who granted me permission to collect data as citizen scientists. I have also managed a citizen science project where in addition to 100+ households granting me access to their yards, they also follow a protocol to monitor wildlife and submit data. Both roles are vital for understanding how wildlife responds to different urban green space management, and my distinction recognizes different levels of engagement. And what data did the 8 citizen scientists collect? Again, unless you are asking specific questions to these 8 citizen scientists about their experiences (and I caution against this due to small sample sizes), it is unclear how their participation enhanced their understanding of conservation, etc. Plus, the survey only sampled those who participated in the No Mow program, and did not conduct a pre-survey. Thus, we don’t know whether their perceptions differ from mowers, and whether their perceptions were formed because of their participation in the program.
L274-277: Can you be more specific? Are there plans to present results to the city planners?
L281-294: Please provide some citations to back up these statements. Also, I suggest a more cautious approach given the small sample size and the comparisons between yards and parks. The limitations of the study should be discussed as per my comment in the method section. Could the results observed be explained because parks don’t have as many planted flowers? What about the yards next to the No Mow lawns? Did they have pollinator gardens or other flowers that might have attracted more bees to the area?
L307-309: I suggest citing Larson et al. 2016
L309-312: I think this is an interesting conclusion and I recommend considering how aesthetics plays into the outreach and awareness. One suggestion is to have future research assess the aesthetic perceptions of the No Mow lawns compared with the mowed lawns. See Ramer et al. 2019.
Table 1: Why only include the most common plant species?
Table 2: I suggest having two ‘Count’ columns: 1 for yards and 1 for parks.
Figure 1: I like that you included all the No Mow sites. However, I think it a little distracting from the bee richness results. I suggest the sites where you didn’t collect bees have a smaller dot. And I presume the green polygons are parks? If so, then it looks like some parks had No Mow??

Suggested additional references to include:
Baldock, K. C. R., M. A. Goddard, D. M. Hicks, W. E. Kunin, N. Mitschunas, H. Morse, L. M. Osgathorpe, S. G. Potts, K. M. Robertson, A. V. Scott, P. P. A. Staniczenko, G. N. Stone, I. P. Vaughan, and J. Memmott. 2019. A systems approach reveals urban pollinator hotspots and conservation opportunities. Nature Ecology & Evolution 3:363.
Fetridge, E. D., J. S. Ascher, and G. A. Langellotto. 2008. The bee fauna of residential gardens in a suburb of New York City (hymenoptera: apoidea). Annals of the Entomological Society of America 101:1067–1077.
Frankie, G. W., R. W. Thorp, M. Schindler, J. Hernandez, B. Ertter, and M. Rizzardi. 2005. Ecological patterns of bees and their host ornamental flowers in two northern California cities. Journal of the Kansas Entomological Society:227–246.
Larson, K. L., K. C. Nelson, S. R. Samples, S. J. Hall, N. Bettez, J. Cavender-Bares, P. M. Groffman, M. Grove, J. B. Heffernan, S. E. Hobbie, J. Learned, J. L. Morse, C. Neill, L. A. Ogden, J. O’Neil-Dunne, D. E. Pataki, C. Polsky, R. R. Chowdhury, M. Steele, and T. L. E. Trammell. 2016. Ecosystem services in managing residential landscapes: priorities, value dimensions, and cross-regional patterns. Urban Ecosystems 19:95–113.
Lowenstein, D. M., K. C. Matteson, and E. S. Minor. 2015. Diversity of wild bees supports pollination services in an urbanized landscape. Oecologia:1–11.
Ramer, H., and K. C. Nelson. 2020. Applying ‘action situation’ concepts to public land managers’ perceptions of flowering bee lawns in urban parks. Urban Forestry & Urban Greening 53:126711.
Ramer, H., K. C. Nelson, M. Spivak, E. Watkins, J. Wolfin, and M. Pulscher. 2019. Exploring park visitor perceptions of ‘flowering bee lawns’ in neighborhood parks in Minneapolis, MN, US. Landscape and Urban Planning 189:117–128.

Reviewer 2 ·

Basic reporting

• The manuscript is written in clear English and is relevant to its hypotheses. Raw data is shared. Figures & tables are provided although edits are suggested.

Other comments:
• Figures and tables should be numbered consecutively based on first discussion in the text (e.g. Table 3 is mentioned before Table 1 and 2; Figure 2 is mentioned before Figure 1). Also, consistency in the capitalization of the first word of the table/figure title is needed, and all abbreviations should be spelled out (e.g. AIC), and in-text references to tables/figures should also be capitalized consistently
• Table 3 isn’t needed – place the AIC values and predictors in the text; you are not presenting multiple variables as you suggest in the text
• Make sure your table and figure legends are fully descriptive. E.g. Figure 1 – should indicate black dots are participating homes and colours represent both areas that were surveyed and the resulting diversity levels in a heatmap. Sample size of houses could be shown. Bee abundances could be presented as well – why aren’t they? A different symbol for urban lawns vs greenspaces would be helpful as well.
• And for Figure 2, is there a standard error or 95% confidence interval? Are there significant differences? This can be shown on the graph
• I’m not sure why you have the “NoMowMay Summary of Results” website page included here; it’s not clear if it’s a copy of your program results website, or trying to show the stats results of your analyses, or what.

Experimental design

• The paper is based on primary research with relevant questions, and is situated in the literature (although very few papers in the literature are referenced). Ethics was obtained for the relevant research. But a lot of the methods and the overall experimental design is missing or unclear. I am also concerned about the comparison of non-mowed lawns to mowed greenspaces in parks, particularly without accounting for surrounding habitat.
• Additional comments:
• I suggest including a definition of citizen science, and more clearly explaining how the citizen scientists were involved. I would not consider No Mow May itself to be a citizen science initiative if people are just taking an action but not collecting and submitting data (i.e. not participating in ‘science’). Volunteers helping to collect data would be citizen scientists but not “regular” home owners. Ditto for survey respondents. You don’t actually try to describe this until line 265. There are a number of papers on this in the literature.
• Line 109: how were the homes and greenspaces selected? Also, why did you choose to compare mowed greenspaces vs lawns? Why not lawns that participated vs did not participate? Having previously surveyed greenspaces is not a relevant reason in my opinion as you do not have previous information for the residential properties for comparison purposes. I’m sure you could get permission from non-participating homes if you asked. you could ask property owners for permission (these could be lawns side by side or on the same street/same community)? It’s not a true comparison if you’re comparing urban lawns to actively managed (mowed & treated with pesticides) parks. Also, I see from the map in Figure 1 that some greenspace/park sites were quite close to each other, while there are parks surrounded by participating homes quite a distance away that were not selected.
• How many homes were within the same 1km of the park/each other? Within 100m of each other? (Line 136 mentions the latter; it should be moved up to the site selection section) Within 100m of the park? I plotted the first three sites in your data and one was across the road from the park (so less than say 20m), another one maybe 70m away, another 150m or so; that suggests that all could be in the same foraging range of bees, and you’re double counting. Alternatively, if you want to say you’re sampling the same population of bees at all sites, so that the population/source of the bees is the same, state that.
• Line 111: who were the citizen scientists? What was their background? What training was provided? Were the authors/researchers out with them? Did all 8 help survey at all sites? Did experts verify their findings?
• Line 112: how were the plants identified? Personal knowledge, a specific identification key?
• Line 113: were they truly randomly selected points? E.g. you put a grid overtop the whole area and used a random number generator to select grids to sample?
• Did you account for other vegetation in the area? There could be a big difference in visitation to lawns that are surrounded by flowering plants in formal gardens/naturalized sections of the property vs those that have only lawn.
• Line 120: how did you come up with 150m2 for the size of park space to sample?
• Line 120: The justification of 100m apart for grids should be referenced, and even then there is ample evidence that bees (even small solitary ones) will forage for more than 100m away from nest sites, so you could still recapture bees in nearby plots. 500m would have been a better option.
• Line 126: so did you survey the entire “no mow” lawn area of each property? Did you survey the entire mowed greenspace area of the parks?
• Line 127: why only one survey period per property? What was the time period for surveys? Some species are only active at certain times of the day. How many people surveyed at one time? Did you only survey/collect bees that were actively visiting flowers on the lawn area vs flying by or visiting flowers that were in other locations (e.g. flower beds, planters)?
• Line 142/survey: some of your questions are leading and do not take into account historic practices. E.g. perhaps a home owner previously only mowed 2x/month – they do not have this option, or even the option to select ‘mow less frequently’. They also didn’t have the option to select ‘no change’ – not everyone may have enjoyed the results or perhaps they will not have to change their practices. Similarly, for changes post-May, they may be already doing some of these actions, which can not be captured by the survey; either they check that they ‘will’ do something based on their participation (which wouldn’t actually be based on their participation) or they do not check anything and that information is lost. Also, did you try to clarify “yard” vs “lawn”? I.e. differentiating between flowers growing in e.g. flower beds vs in the lawn?
• Survey vs summary of results (external files): so were there over 500 households participating or 435? Or were over 500 homes including those deemed by the researchers to be participating but not registered (and how did they determine they were participating in the initiative vs just not mowing for whatever reason).
• Line 152: was abundance based on all species combined?
• Is the effect due to the urban garden resources vs greenspace resources?
• Was there a control in the survey to account for those who already managed habitat for pollinators, etc? Did properties with more gardens have more pollinators regardless as to how much lawn was not mowed? Were properties with more gardens already established more likely to participate in No Mow May than those with no/fewer gardens?
• Discuss what plants were flowering in the lawns – native vs non-native? Preference for some species of plants over others? Do you consider not cutting the grass to be the same as naturalizing?

Validity of the findings

• The analyses were appropriate, but the underlying comparison of lawns to greenspaces is not necessarily appropriate or relevant. Some analyses that could have been conducted were not or results were not clearly presented. This may be best treated as a pilot study.
• Additional comments:
• Line 195/Summary of results (external document): Do you have any “proof” (even personal communications with residents) that all of this “protected” “pollinator habitat” was only due to No Mow May (vs would have existed anyway)(i.e. do some residents normally not mow in May, or is based on surrounding vegetation)?
• Line 195/Summary of results (external document): were you including the mowed greenspaces in your total amount of habitat being protected by the initiative (because they did not change management practices)?
• Line 165: please indicate here if the list of species and abundances are being provided in a table
• Line 175: you refer to “common and abundant” – what is commonness based on? Abundance based on?
• Table 1: provide percent coverage for each species if you have that data, or percent of plots, or some measure of abundance and cover/forage; it is possible for one species to fill 100% of a plot or only 5% of a plot; ,
• Table 1: why not include all plant species here? If the list is too extensive, why not include in a supplementary table?
• Table 1, 2: provide the scientific authorities for each species
• Table 2: provide relative (percent) abundance for each species for easy comparison in addition to counts, and if they were present in one or both of lawns & greenspaces
• Line 182: provide model results here, including model statistic, df, p-value (this is not presented in paper currently)
• Line 199: provide actual results of the survey in a table (in-text or supplementary)
• Line 265: it’s not a city wide experiment if you aren’t actually testing across the city; you focused intently in just a couple areas
• Provide discussion about what plant species provide the most forage
• Discuss if impacts on pollinators would be consistent or change over the course of the month.
• Discuss the impacts of lawns being mowed after May ends (i.e. did home owners create a sink by attracting and supporting pollinators and then removing that food?)
• Discuss the value of lawns and other areas being mowed occasionally to create a new flush of growth vs never being mowed.

Additional comments

• Line 50: I disagree that we need to protect non-native species, particularly in urban areas that may be a refuge for native ones. Please supply references to support your claim that these species need to be protected. You later (e.g. line 63) focus on native bees. Native bees can provide ecosystem services, and in many cases can be better than honeybees. Honeybees are a managed agricultural farm animal like cattle, chicken, etc. and it is important that researchers and the public understand the distinction between native and commercially produced species.
• Line 66: remove the word “the” in “the late April”
• Line 74,77: add a reference for the initiatives (even a URL if no reports, etc. available)
• Line 104: add a comma after the organization name
• Line 174: repeated “mowed areas” three times; remove repeated words and replace with correct term
• Line 174: add in “plant” or “flower” species here if that’s what you mean (vs bee species)
• Line 247: remove the comma after Appleton
• Line 250: who provided the educational opportunities? What type of reach did you get? Relatedly, Line 269: how did you improve the general public’s understanding of “key ecological and conservation issues” and “understanding of urban ecosystems” – did you discuss all of these items on your Facebook page? Did you do any media outreach? Was the messaging just about pollinators or were there others?
• Line 270: were all >400 households invited to help collect data? Were only 8 people from across the city interested?
• Line 281: why are lawns easily accessible? By whom? Does that mean forests/meadows/wetlands are not accessible?
• Line 320: who are the 3 citizen scientists not mentioned here? At least say ‘and 3 anonymous individuals’ to acknowledge them if they don’t want their names listed.
• Line 326: remove the word “a” before registered

·

Basic reporting

Overall, the manuscript is well written, well referenced and thoughtfully structured. The motivation, objectives and methods are generally clear, with plenty of context to place. There are a few areas that could be re-written to improve readability and comprehension; or made more concise - I have identified these in the pdf annotation.

1. The objectives included the citizen science component (i.e. how to engage and include citizens within a community programme such as No Mow May), which is an important consideration. However, it seemed at times that the process of the citizen science component overwhelmed the science component of the paper (i.e. details of council meetings and administrative process in the Methods, and parts of the Discussion). I feel like this paper would be stronger with some of these details made more concise, or included as supplementary material. I have included more detail in the PDF annotation.

2. There is a tendency of the paper to be USA focused, which may render it less relevant to international readers. Include a few more specific details of the study area (i.e. location relative to the contiguous USA on Fig 1., population, etc.). I have annotated the PDF with a few more notes.

3. The figures and the text could benefit from more consistency in the terminology between the two treatments. Can I suggest 'Unmown' and 'Regularly Mown' to identify them.

4. I thought the referencing was a little incomplete, given the few studies on lawn management on insects generally, and bees specifically. While many studies discuss other insect groups, e.g. Hemipterans, Scarabideae, these could be interesting to discuss in order to strengthen the central argument of the paper. A summary of these papers is presented in one of my research group's meta-analyses: Watson et al. Journal of Applied Ecology 2019. DOI: 10.1111/1365-2664.13542

In particular, the following is very relevant and should be included: Wastian, L., Unterweger, P. A., & Betz, O. (2016). Influence of the reduction of urban lawn mowing on wild bee diversity (Hymenoptera,
Apoidea). Journal of Hymenoptera Research, 49, 51–63. https://doi.
org/10.3897/JHR.49.7929

Experimental design

The research question is clearly stated and very relevant within the context of current urban greenspace management. The methods are described reasonably well, but a few details were lacking and/or confusing (see below for more specific details). This research certainly fills a knowledge gap and this is well identified. The number of replicates seems a little low given the availability of

1. From an experimental design perspective, there is a minor problem with comparing residential yards and parks/sporting fields as the two treatments: the variety of maintenance activities in residential lawns can be quite different, whereas public lands tend to be more consistently managed but have more aggressive management, such as widespread pesticide treatment, shorter cuts, etc. While this does not invalidate the results, I think it is worth addressing to identify this potential to the reader.

2. I had a little difficulty following some of the methods, and identifying which variables were collected. I suggest a review of the methods section to clearly identify what was collected and how this was done. In particular, the use of '% of unmown surface' as a primary variable was not clearly defined. Is this simply a measure of the surface area of each yard, or the proportion of unmown relative to mown? This is a critical element and needs to be clarified, then discussed in the Discussion.

3. I found the results and explanation of the post-project survey a little confusing. Specifically, many of the questions are framed as yes/no, and it is not explained that respondents have a scale on which to respond. The results of the survey (certain questions) may better be presented as a figure to clearly show how participants responded on this scale.

4. The floral diversity data could likewise benefit from a figure to clearly show the results.

Validity of the findings

The findings of the study (most notably, the bee and plant diversity components) are well designed and conducted. The data has been provided, and the statistical methods presented. The conclusions draw together the scientific research and the citizen science findings, though for mine the citizen science discussion dominates the importance of the biodiversity findings.

1. I find the use of the models a little superfluous, given that there are few input variables, and limited discussion of why these were selected, covariation, etc. The use of AIC in this case is irrelevant as the AIC is only useful to compare the fit of models, so should not be presented in this context. Perhaps reconsider if these models are necessary, or change the way they are presented and discussed.

Additional comments

Thank you for a welcome study that supports a surprisingly small volume of research in this field. The results of the bee diversity data were expected, but welcome - particularly to see the magnitude of increase in unmown areas. This is really important work, and important to publish.

While I understand the motivation for wanting to include lots of detail on the citizen science component, I would encourage reducing the detail of this a little. At times it overwhelmes the bee and floral data. Likewise, some parts of the methods and results are rendered a little confusing which detracts from the overall strong results and thoughtful message of the paper - I would suggest a thorough revision of the language of the manuscript to ensure it thoroughly explains the methods and variables collected, and that the messages and arguments are presented strongly and concisely. Picture this as being read by a global audience because it certainly will be!

Thanks again. I am eager to see more work from this research group and this part of the world.

---

## Round 0.2 · Minor Revisions

Thank you for addressing the previous round of reviewer comments. I believe the most pressing issues have been dealt with. I would be happy to accept your paper if you could first address these minor changes requested by Reviewer 2.

Thank you again.

Reviewer 2 ·

Basic reporting

• I think the authors did a good job in responding to the earlier criticism.
• I find Figure 1 hard to read and understand
o you indicate that all black points/dots are no mow sites; presumably you mean all sites that had registered for No Mow May not all study sites; while I realize you probably want to highlight how involved the city was in in participating in the intiatiive, showing the other points is not applicable here and I would remove them; if you’re keeping them in, adjust the legend to clarify that the black dots are all registered households; perhaps change them to be a different symbol (e.g. star, bee, flower, etc.)
o the size of the larger coloured circles is much larger than that of the black dots; if the coloured circle is a now mow study site, why is it is so much bigger than the dots? Which of the dots were selected, when there are multiple per circle?
o You can also crop the map in a lot more, to just surround the areas that were studied vs all of the outlying city area (if people want to see what the city looks like, they can google it); this will help make the points more visible, as right now they are overlapped and hard to distinguish (this would almost double the visible size of the study area)
• Table 1: I still think you could list all plant species in table 1, otherwise indicate that these are the top x of y species found, or just the species that were found in >1 property, etc..
• There is no reference in-text to supplemental materials, yet in the author rebuttal letter they reference supplemental materials for plant lists, survey results, and model output. Will the authors be providing the entire Excel file as a supplemental file? I suggest separating out and further summarizing.

Experimental design

• In their rebuttal, the authors indicate that the comparison of mowed greenspaces to unmowed lawns is appropriate as they have done surveys of various vegetation types, which is being presented in other papers; I still do not agree with this, but understand that they did not get enough permissions to conduct surveys in mowed areas. This should be explicitly discussed in the methods and discussion, and the possible repercussions/influences on the data. The fact that the parks have been managed consistently the same way for decades does not apply here, as you don’t have the baseline data for the mowed vs unmowed sites. I.e. expand the discussion more about this.
• The authors indicate in their rebuttal and line 192 in the track-changed text that “every effort was made to avoid double counting and recapturing individuals.” But didn’t explain how they did this. Additionally, in the text (line 117 of track changed file) “Homes were selected based on remaining not mowed through the entire month, and their proximity (within 1 km) to the focal parks within each of the five neighborhoods. Homes were separated by at least 100 meters to reduce the likelihood of resampling individual bees at multiple homes.” but with sites only 100m apart according to your text (and in some cases, across the road and other locations <100m away, in my plotting of points), and with some specimens being released at the end of the individual household survey period, you’re still sampling the same population of bees, so the sites aren’t independent
o Were you trying to have all no-mow May sites and mowed parks be withing the same 1km radius so they are sampling the same bee population? In that case, the no-mow sites should be considered sub-sites/samples – clarify more.
o In the author rebuttal it sounds like you didn’t want to collapse sites/subsites in order to keep power (i.e. “As for independence of samples, a spatial autocorrelation test with three points may not be statistically appropriate here. We got around the independence issue by sampling multiple parks around town with distances typically >2 km apart from each other. The three plots in each park show within park variability, by lumping them into a single metric our sample size and power would decrease from n=15 to n=5“; the desire to keep power/sample size is not a valid reason on its own to ignore independence though
o Line 192 should be clarified to explain how you avoided recollecting the same bees; indeed, also discussion should be included about this and if there was any impact on lethally collecting bees from within the same study area as other homes and parks in terms of bee abundance in later surveyed areas

Validity of the findings

• While I still have concern about the comparison of mowed parks to unmowed residential areas, the findings are of interest and valid with the expanded discussion indicated above.

Additional comments

• There are still references to citizen/community science in this study that I believe should be removed; e.g. first line of abstract (“No Mow May is a community science initiative”) and line 131 of track changed file (“Working with city government and community scientists”): As you addressed in your overall rebuttal comments, you agreed that the act of not-mowing lawns is not a community science participation on its own. You can say it’s a community initiative, or working with community members, or similar, but not community science. Your discussion still brings in community science (Line 360 (having just a couple volunteers assisting researchers isn’t really the same thing as a community science program)
o Line 365, 369, etc.: be consistent in use of community science vs citizen science
o Line 424, 425, etc.: be consistent in citizens vs community members
• Line 55 of tracked changed file: urban areas provide nesting and overwintering sites in addition to foraging sites; even if the emphasis of your paper is on foraging, I would include these other habitat types that can be provide here as well
• Line 57: why specifically list honeybees here as an example species and not list examples of the other native or non-native bees? The way you list honeybees as the sole example seems to give them more weight or importance.
• Line 125, 163, 236,256, 399etc: be consistent in capitalization (and terminology) of No Mow May, No Mow Area (and un mowed vs not mowed)
• Line 200: this implies that all 435 homes participated in the survey yet earlier (Line 124) you indicated only 130 did, a 30% response; I’d either move the response rate to this section, or report based on the 130 responses you did get
• Line 384: give a few examples of these other drivers (e.g. as reviewers had pointed out)

---

## Round 0.3 · accepted · Accept

Thank you for addressing the issues raised by reviewers. I am happy to accept your manuscript at this point.